# In Vitro Antimicrobial Activity of the Siderophore Cephalosporin Cefiderocol against *Acinetobacter baumannii* Strains Recovered from Clinical Samples

**DOI:** 10.3390/antibiotics10111309

**Published:** 2021-10-27

**Authors:** Davide Carcione, Claudia Siracusa, Adela Sulejmani, Roberta Migliavacca, Alessandra Mercato, Aurora Piazza, Luigi Principe, Nicola Clementi, Nicasio Mancini, Valerio Leoni, Jari Intra

**Affiliations:** 1Department of Laboratory Medicine, University of Milano-Bicocca, Azienda Socio Sanitaria Territoriale Brianza ASST-Brianza, Desio Hospital, via Mazzini 1, 20833 Desio, Italy; davide.carcione@gmail.com (D.C.); claudia.siracusa@asst-brianza.it (C.S.); a.sulejmani@campus.unimib.it (A.S.); valerio.leoni@asst-brianza.it (V.L.); 2Unit of Microbiology and Clinical Microbiology, Department of Clinical-Surgical, Diagnostic and Pediatric Sciences, University of Pavia, 27100 Pavia, Italy; roberta.migliavacca@unipv.it (R.M.); alessandra.mercato@unipv.it (A.M.); aurora.piazza@unipv.it (A.P.); 3Clinical Pathology and Microbiology Unit, S. Giovanni di Dio Hospital, 88900 Crotone, Italy; 4Laboratorio di Microbiologia e Virologia, Università Vita-Salute San Raffaele, 20132 Milan, Italy; clementi.nicola@hsr.it (N.C.); mancini.nicasio@hsr.it (N.M.); 5IRCCS Ospedale San Raffaele, 20132 Milan, Italy; 6Clinical Chemistry Laboratory, University of Milano-Bicocca, Azienda Socio Sanitaria Territoriale di Monza ASST-Monza, San Gerardo Hospital, via Pergolesi 33, 20900 Monza, Italy; j.intra@asst-monza.it

**Keywords:** antimicrobial resistance, nosocomial infection, cefiderocol, siderophore-cephalosporin, *Acinetobacter baumannii*

## Abstract

Background: Cefiderocol is a siderophore cephalosporin that exhibits antimicrobial activity against most multi-drug resistant Gram-negative bacteria, including Enterobacterales, *Pseudomonas aeruginosa*, *Acinetobacter baumannii*, and *Stenotrophomonas maltophilia*. Methods: A total of 20 multidrug-resistant *A. baumannii* strains were isolated from 2020 to 2021, molecularly characterized and tested to assess the in vitro antibacterial activity of cefiderocol. Thirteen strains were carbapenem-hydrolysing oxacillinase OXA-23-like producers, while seven were non-OXA-23-like producers. Minimum inhibitory concentrations (MICs) were determined by broth microdilution, considered as the gold standard method. Disk diffusion test was also carried out using iron-depleted CAMHB plates for cefiderocol. Results: Cefiderocol MICs ranged from 0.5 to 1 mg/L for OXA-23-like non-producing *A. baumannii* strains and from 0.25 to >32 mg/L for OXA-23-like producers, using the broth microdilution method. Cefiderocol MIC_90_ was 8 mg/L. Diameter of inhibition zone of cefiderocol ranged from 18 to 25 mm for OXA-23-like non-producers and from 15 to 36 mm for OXA-23-like producers, using the diffusion disk method. A large variability and a low reproducibility were observed during the determination of diameter inhibition zone. Molecular characterization showed that all isolates presented the ISAba1 genetic element upstream the *bla*OXA-51. Among OXA-23-like non-producers, four were *bla*OXA-58 positive and two were negative for all the resistance determinants analyzed. Conclusions: Cefiderocol showed in vitro antimicrobial activity against both carbapenem-susceptible and non-susceptible *A. baumannii* strains, although some OXA-23-like producers were resistant. Further clinical studies are needed to consolidate the role of cefiderocol as an antibiotic against MDR *A. baumannii*.

## 1. Introduction

Species belonging to the *Acinetobacter* genus are strictly-aerobic, non-fermentative Gram-negative coccobacilli, predominantly found in water, soil, and sewage. Over 50 different species of *Acinetobacter* genus are known, but only a few species, belonging to the *A. calcoaceticus-baumannii* complex, cause the majority of community and healthcare associated infections [1,2,3]. *A. baumannii* is considered an opportunistic pathogen, and risk factor for developing an infection include mechanical ventilation, prolonged hospitalization, immune suppression, comorbidities, major trauma or burns, previous antibiotic use, invasive procedures, and presence of urinary catheters [1,2,3,4]. *A. baumannii* can be transmitted via aerosol droplets, person-to-person contact, skin, sputum, urine, and feces [1,2]. Several infections such as pneumonia, bacteremia, catheter-related and urinary tract infections, skin and soft tissues infections, and meningitis are caused by this microorganism.

Listed as an ESKAPE (*Enterococcus faecium*, *Staphylococcus aureus*, *Klebsiella pneumoniae*, *Acinetobacter baumannii*, *Pseudomonas aeruginosa*, and *Enterobacter spp.*) nosocomial pathogen, carbapenem-resistant A. *baumannii* is classified by the World Health Organization as a pathogen for which development of new treatments are urgently needed [5,6]. In the 2019 Antibiotic-Resistance Threats Report, the Center of Disease and Control recategorized A. *baumannii* from hazard level “Serious” to hazard level “Urgent”, meaning that the level of antimicrobial resistance needs more aggressive actions [1,2]. Mortality rates from MDR Acinetobacter range from 23 to 68%, even though it is difficult to obtain an exact attributable mortality rate due to poor prognosis in patients already featuring several comorbidities [7,8].

Limited treatment options are available in infections caused by MDR *A. baumannii*. The knowledge of the pathogenesis, the virulence factors, and antibiotic resistance mechanisms of *A. baumannii* such as β-lactamase acquisition, up-regulation of drug efflux pumps, aminoglycoside modification, permeability defects, and alteration of target sites [9], are important to improve the management of infections caused by this microorganism.

*A. baumannii* is intrinsically resistant to several commonly used antibiotics, including aminopenicillins and cephalosporins. It also presents a remarkable ability to acquire mechanisms that confer resistance to aminoglycosides, fluoroquinolones, tetracyclines, and carbapenems. As a consequence, *A. baumannii* carbapenem resistance represents one of the main concerns. Furthermore, resistance to polymyxins and tigecycline has also been reported, thus indicating that *A. baumannii* can be fully resistant to current available antimicrobials [10].

*A. baumannii* carbapenem resistance is conferred by different mechanisms, including decreased outer membrane permeability, β-lactamase production, and modification of penicillin-binding proteins. The most common mechanism of carbapenem resistance is due to carbapenem-hydrolysis enzymes that belong to Ambler’s class D and B β-lactamases. Resistance genes may be located both on the chromosome and on mobile genetic elements (i.e., integrons, transposons, and plasmids). Currently available methods to detect carbapenem-resistance differ in accuracy and efficiency [11]. Therefore, the implementation of different approaches, based on mass spectrometry and molecular methods, is ongoing [11]. Amongst currently available therapeutic options, polymyxins, such as colistin and polymyxin B, are the antibiotics presenting the highest level of in vitro activity against MDR *A. baumannii* [12]. Although polymyxin B seems to be related to lower renal toxicity compared to colistin, the latter is widely used in clinical practice. Combination therapy, in contrast to monotherapy, seems to increase microbiological eradication rates [13], but evaluations of novel treatment options for infections caused by MDR *A. baumannii* are needed. Among the new approved antibacterial drugs against Carbapenem-Resistant Enterobacteriaceae (CRE), only one, cefiderocol, a siderophore cephalosporin that takes advantage of iron uptake mechanisms to facilitate cell entry, is endowed with broad-spectrum activity against carbapenem-resistant *A. baumannii* spp., *P. aeruginosa*, and *Stenotrophomonas maltophilia*.

Cefiderocol (formerly named S-649266) is a combination of a catechol-type siderophore and a cephalosporin core with side chains similar to ceftazidime and cefepime, which are third- and fourth-generation cephalosporins, respectively [14]. A catechol moiety on the 3-position of the R2 side chain allows cefiderocol to function as a siderophore molecule chelating extracellular iron. Following the chelation of iron, cefiderocol is transported to the periplasmic space through ferric iron transport systems located on the outer membrane of Gram-negative bacteria. Once within the periplasmic space, cefiderocol dissociates from the iron and binds to penicillin-binding proteins (PBP), inhibiting peptidoglycan cell wall synthesis [14]. This mechanism of action allows high intracellular penetration into the periplasmic space. Importantly, cefiderocol is resistant to the hydrolysis by β-lactamases, including extended spectrum β-lactamases, such as CTX-M, and carbapenemases, such as KPC, NDM, VIM, IMP, OXA-23, OXA-48-like, OXA-51-like, and OXA-58-like [15,16,17]. The FDA approved the cefiderocol for the treatment of both complicated urinary tract infections (UTIs) in 2019 and hospital-acquired bacterial pneumonia, including ventilator-associated bacterial pneumonia, in 2020 [18].

In light of the importance of cefiderocol as a therapeutic option, the main purpose of this work is to assess in vitro antibacterial activity of cefiderocol against MDR *A. baumannii* strains isolated from clinical samples collected at the Microbiology Laboratory of Desio Hospital, Italy, and at the Microbiology and Molecular Microbiology Laboratory of the University of Pavia.

## 2. Results

Out of the 20 *A. baumannii* strains, 13 isolates were carbapenem-hydrolysing oxacillinase OXA-23-like producers, while seven were OXA-23-like non-producers. Table 1 shows the MICs distributions of the antibiotics tested for the clinical isolates included in this study [19]. All isolates were confirmed as MDR, with 19/20 (95%) being resistant to ciprofloxacin, 19/20 (95%) to levofloxacin, 19/20 (95%) to trimetoprim-sulfametoxazol, 18/20 (90%) to amikacin, 19/20 (95%) to gentamicin, and 19/20 (95%) to meropenem. Only the AB 14C04 isolate was susceptible to ciprofloxacin, trimetoprim-sulfametoxazol, amikacin, levofloxacin, and gentamicin. The AB 2MG isolate was susceptible to meropenem. All isolates were susceptible to colistin with MICs ranging from <0.5 to 2 mg/L. Although there is insufficient evidence that *A. baumannii* is a good target for therapy with tigecycline and minocycline [19], these antibiotics showed antimicrobial activity with MICs < 0.5 mg/L and MICs < 4 mg/L, respectively, except for two isolates, AB5968 and AB 9063, which resulted in being resistant to minocycline. Cefiderocol MICs ranged from 0.25 to >32 mg/L for all isolates, particularly from 0.5 to 1 mg/L for OXA-23-like non-producers and from 0.25 to >32 mg/L for OXA-23-like producers, using the broth microdilution method. Diameter of inhibition-zone of cefiderocol ranged from 15 to 36 mm, particularly from 18 to 25 mm for OXA-23-like non-producers and from 15 to 36 mm for OXA-23-like producers, using the diffusion disk method. However, a large variability and a low reproducibility were observed in the determination of diameter inhibition-zone. Figure 1 shows the antimicrobial activity of cefiderocol by comparing the MIC distribution and the zone diameter distribution: a low concordance was observed.

Using the European Committee on Antimicrobial Susceptibility Testing (EUCAST) clinical PK/PD breakpoints (sensitive: ≤2 mg/L, using broth microdilution method; sensitive: ≥17 mm, using disk diffusion technique), among OXA-23-like non-producers 5/7 (71%) isolates were susceptible to cefiderocol and 2/7 (29%) resistant, using broth microdilution method; conversely, 100% of strains were within sensitivity using disk diffusion technique. On the other hand, among OXA-23-like producers, 5/13 (38%) isolates were susceptible to cefiderocol and 8/13 (62%) resistant using broth microdilution method; on the contrary, using disk diffusion technique, 19/20 (95%) strains were sensitive and 1/20 (5%) resistant. The concentration of cefiderocol inhibiting 90% of isolates tested (MIC_90_) was 8 mg/L, particularly 1 mg/L for OXA-23-like non-producers and 8 mg/L for OXA-23-like producers, using the broth microdilution method. On the other hand, a zone diameter of 19 mm inhibited 90% of isolates tested using disk diffusion technique.

The results of molecular characterization are reported in Table 2. Among the seven *bla* OXA-23-like-negative strains, two (AB 11–69, AB 2 MG) resulted negative for all the resistance determinants investigated; four were *bla*OXA-58-like positive. All isolates showed the presence of the ISAba1 genetic element upstream the *bla*OXA-51-like gene. Aminoglycoside resistance determinants *aph6A* and *armA* have been detected in seven and eight strains, respectively, with one case of co-presence. Eleven different clones have been identified by PFGE (Figure 2).

The clone A, which also includes two subclones, emerged in 2003 and was also identified in three different hospitals in 2011 and 2014. The most represented clone was the clone G, comprising two subclones, detected in 2012 at first and then isolated in the same hospital for the following two years. The Italian SMAL clone was detected in two hospitals in 2003 and 2009. None of the bacterial isolates were clonally related to the two international clones ICL-I and ICL-II by PFGE. The MLST analysis highlighted the presence of 10 different STs (Table 2). Instead of the PFGE results, the majority of the isolates (*n* = 8) belonged to the international clone ST2; two strains to the ST78. The others belonged to the STs 4, 10, 19, 109, 2681, 1077. Furthermore, two new STs (not yet presented in the Pasteur database) have been identified, thus showing a great clonal heterogeneity in the isolates.

## 3. Discussion

The increasing number of MDR *A. baumannii,* among other MDR pathogens, has forced the development of antibiotics endowed with novel mechanisms of action against carbapenem-resistant Gram-negative bacteria. Cefiderocol, a siderophore cephalosporin approved in 2019 for the treatment of complicated UTIs, represents a suitable option against MDR infections, under the light of the few therapeutic treatments available [20,21,22,23,24].

This study evaluated the in vitro activity of cefiderocol against 20 *A. baumannii* strains isolated from clinical specimens by analyzing data from different methods routinely used by microbiology laboratories to determine the antimicrobial susceptibility of bacteria: the microdilution method, which is considered the gold standard [25], and the disk diffusion test. 

The microdilution broth method, as well as the Etest, determine MIC values that can be used by clinicians to address antibiotic therapy options; whereas, the disk diffusion is a qualitative test based on a diameter of inhibition zone, which cannot be easily converted into a MIC value, but only allowing to describe bacteria as “drug susceptible”, “intermediate” and “drug resistant”. Although simple and performed in many laboratories, the disk diffusion assay presents some limitations including low reproducibility and inaccurate inhibition diameter determination [26,27,28,29]. The results from disk diffusion test performed for cefiderocol on our MDR *A. baumannii* clinical isolates, were not fully coherent with data obtained by using the “gold standard” broth dilution method due to the variability observed amongst technical replicates. Recently, Morris and coauthors compared cefiderocol disk diffusion method to broth dilution on carbapenem-resistant Enterobacterales and non-glucose-fermenting Gram-negative bacilli. They observed that disk diffusion is a convenient alternative approach to broth dilution for cefiderocol antimicrobial susceptibility testing, except for *A. baumannii* complex isolates [30]. Moreover, Albano and coauthors determined the MICs of cefiderocol for 610 Gram-negative bacilli. Broth dilution and agar dilution methods were used. The results showed significant discordance between agar dilution and broth dilution. Again, broth dilution proved to be the most reliable method for determining cefiderocol MICs [31]. 

For these reasons, our analysis focused on results obtained from the microdilution broth method.

Cefiderocol showed strong antibacterial activity, with MIC_90_ values of ≤8 mg/L for all isolates, and in particular the MIC_90_ was eight times lower for OXA-23-like non-producers. On the other hand, among OXA-23-like producers, five of our *A. baumannii* strains were resistant to cefiderocol (>16 mg/L). Our data are consistent with previous studies. Delgado-Valverde and coauthors found an MIC_90_ value of 4 mg/L, with 95% of isolates sensitive to cefiderocol [32]. Different works reported MICs of > 8 mg/L for *A. baumannii* isolates, while others have observed *A. baumannii* strains susceptible to cefiderocol, with MICs values of 0.5 mg/L [33,34,35,36,37]. Dobias and couthors found seven OXA-23-like producing *A. baumannii* strains with cefiderocol MICs > 8 mg/L [38].

The outer membrane of Gram-negative bacteria is able to limit the access of antibiotics to the cell targets. *A. baumannii* is characterized by a low outer membrane permeability and a rapid efflux via due to the presence of numerous efflux systems [39]. Two TonB-dependent receptors (TBDRs), named Ab-PiuA and Ab-PirA, have been identified as the major uptake systems for the siderophore-iron complexes [40]. Down-regulation of iron transport receptors in *P. aeruginosa* have been associated with resistance to siderophore-drug conjugates [41]. Recently, Malik and coauthors reported that reduced expression of the siderophore receptor gene *pirA* is correlated with resistance to cefiderocol in *A. baumannii.* Moreover, mutations involving the PBP3 may also contribute to the resistance to cefiderocol [42]. The resistance to cefiderocol observed in our *A. baumannii* strains, particularly among OXA-23-like producers, could be explained by the reduced expression of siderophore receptors and not by β-lactamase activity due to the presence of different oxacillinases. Additional molecular studies will be needed to identify and characterize all TBDRs involved in this uptake mechanism, in order to improve the susceptibility to antibiotic-siderophore complex and to minimize the development of resistance. 

The investigation of clonality among isolates showed a great variability, with several STs and PFGE clones identified. No correlation between resistance phenotypes and particular STs has been identified. The four cefiderocol resistant strains belonged to as many different PFGE clones and three different STs. These findings highlight that the emerging cefiderocol resistance is not a trait associated with a specific genetic lineage and therefore the cefiderocol resistance is not predictable among clonally related *A. baumannii* strains. Moreover, no correlation between type of specimen and clonality lineages was observed.

A multicenter study accounting for a higher number of clinical isolates would be needed in order to improve accuracy of experimental determinations

## 4. Materials and Methods

### 4.1. Organism Identification and Antimicrobial Susceptibility Testing 

A total of twenty A. *baumannii* complex strains, isolated from clinical specimens (bronchoaspirate, *n* = 6; tracheal swabs, *n* = 5; urine, *n* = 4; blood culture, *n* = 2; wound, *n* = 1; peritoneal fluid, *n* = 2) between 2020 and 2021 in the Microbiology Laboratory of Desio Hospital, Italy, and coming from a collection of the Microbiology and Molecular Microbiology Laboratory of the University of Pavia, were well-characterized. The identification of bacteria was performed by matrix-assisted laser desorption/ionization time-of-flight mass spectrometry (Vitek^®^ MALDI-TOF MS). *E. coli* ATCC^®^ 8739 was used as control.

Antimicrobial in vitro activity of cefiderocol was determined by two different approaches: (I) broth microdilution method using Sensititre^TM^ Cefiderocol MIC panel CMP1SHIH (Thermo Fisher, Waltham, MA, USA) following the manufacturer’s instructions; (II) disk diffusion method using 30 µg discs of Cefiderocol (LiofilChem^®^, Roseto degli Abruzzi, Italy). Broth microdilution panels included the following ranges of antimicrobial agents: ciprofloxacin (0.06–1 mg/L), trimetoprim-sulfametoxazol (1–8 mg/L), amikacin (4–16 mg/L), levofloxacin (2–8 mg/L), colistin (0.5–4 mg/L), gentamicin (1–8 mg/L), meropenem (0.12–64 mg/L), ceftazidime/avibactam (1–64 mg/L), ceftolozane/tazobactam (0.5–64 mg/L), (Micronaut-MERLIN Diagnostika GmbH), and cefiderocol (0.03–32 mg/L) (Sensititre Thermofisher). Cefiderocol susceptibility was tested in iron-depleted cation-adjusted Mueller-Hinton broth and plate (ID-CAMHB), while all other antimicrobial agents were tested using standard CAMHB. In parallel, on each day of the testing, *E. coli* ATCC^®^ 25922, *P. aeruginosa* ATCC^®^ 27853, and *K. pneumoniae* ATCC^®^ 2814 were used as control strains, in order to check that all results were within the EUCAST ranges for all antibiotics tested, including cefiderocol [19].

We defined an *A. baumannii* isolate as Multi-Drug Resistant (MDR) if it exhibited a non-susceptibility to at least one agent in three or more antimicrobial categories [43]. Resistant and intermediate resistant *A. baumannii* isolates were combined, as previously reported [19].

### 4.2. Phenotypic Detection of Carbapenemase OXA-23-Like

The OXA-23 K-SeT^®^ immunochromatographic assay (Coris BioConcept, Gembloux, Belgium) was performed to efficiently detect OXA-23-like carbapenemases, without cross-reactions with other OXA carbapenemases, such as OXA-24, OXA-72, OXA-58, OXA-143, OXA-48, and OXA-198, non-acquired OXA carbapenemases or non-carbapenemase OXA [44].

### 4.3. Molecular Characterization of the Isolates

#### 4.3.1. Antibiotic Resistance Genes Investigation

The genomic DNA of the 20 *A. baumannii* isolates was extracted using the automated Puro extraction system (DID, Milan, Italy), with the DNA tissue kit, according to manufacturer’s instructions. PCR and Microarray Check-MDR CT103 XL (Check Points) assays were performed for all the isolates to investigate the presence of antibiotic-resistance genes. In particular, amplification of class D β-lactamase genes (*bla*OXA-51-like, *bla*OXA-58-like, *bla*OXA-24-like, *bla*OXA-23-like), aminoglycosides resistance determinants (*armA* and *aph6A*), Insertion Sequences (IS) elements, class 1 integrons variable regions, were carried out, as previously reported [45,46]. The presence of ISAba1 elements adjacent to blaOXA-51-like and blaOXA-23 genes was also investigated, as previously described [47]. The location of ISAba1 sequence in the upstream regions of these resistance genes may facilitate their overexpression, thus increasing the resistance level to carbapenems [47].

#### 4.3.2. Clonal Relatedness and Typing

##### Pulsed-Field Gel Electrophoresis (PFGE)

Genomic relatedness among *A. baumannii* isolates was investigated by Pulsed-Field Gel Electrophoresis (PFGE). The genomic DNA of the isolates was digested with ApaI restriction enzyme (35 U/sample; Promega Corporation, Madison, WI, USA) and fragments were separated on a CHEF-DR II system (Bio-Rad) at 14 °C for 25 h at 6 V/cm with an initial pulse time of 0.5 s and a final pulse time of 30 s. Lambda 48.5 kb concatemers (New England BioLabs, Beverly, MA, USA) were used as molecular size markers. DNA restriction patterns and the dendrogram of strains relatedness were analyzed and obtained with Fingerprinting II version 3.0 software (Bio-Rad Laboratories, Inc., Hercules, CA, USA) using the Unweighted Pair Group Method with Arithmetic Averages (UPGMA). The Dice correlation coefficient was used with a 1.2% position tolerance. Only bands larger than 40 kb were considered for the analysis. Strains were considered clonally related in the case of >85% similarity [46]. *A. baumannii* RUH875 and RUH134 were used as reference strains representative of the International Clonal Lineages I (ICL I) and II (ICL II).

#### Multilocus Sequence Typing (MLST)

Multilocus sequence typing (MLST) of *A. baumannii* isolates was performed using the seven primers (Pasteur scheme) and conditions described on the Institut Pasteur website (https://pubmlst.org/organisms/acinetobacter-baumannii, accessed on 10 March 2021). The obtained amplicons were purified using the quantum Wizard^®^ SV Gel and PCR Clean-Up System (Promega, Madison, WT, USA) and subjected to bidirectional Sanger sequencing. Sequences were analyzed using the online BLAST web server (http://www.ncbi.nlm.nih.gov/BLAST/, accessed on 10 March 2021) and MultAlin (http://bioinfo.genopole-toulouse.prd.fr/multalin/, accessed on 10 March 2021) software. Analyses of allele sequences and sequence type (ST) assignment were performed using the Oxford *Acinetobacter baumannii* website (http://pubmlst.org/abaumannii/, accessed on 10 March 2021).

## 5. Conclusions

Cefiderocol demonstrated in vitro antimicrobial activity against both carbapenem-susceptible and non-susceptible *A. baumannii* strains, regardless of both the type of carbapenemase present and clonality. The cefiderocol resistance observed in our analysis, might be associated with reduced, downregulated, or absent expression of siderophore receptors, which allow the entry of antibiotics into the bacterial cell. Molecular analysis of genes encoding for siderophore receptors are certainly needed to investigate these drug resistance mechanisms, to better define the role of cefiderocol for the treatment of MDR *A. baumannii* infections. In addition, it could be useful to perform broth dilution assays in the presence of different iron concentrations, with the aim to observe the performance of cefiderocol activity in a simulated host system [48].

## Figures and Tables

**Figure 1 antibiotics-10-01309-f001:**
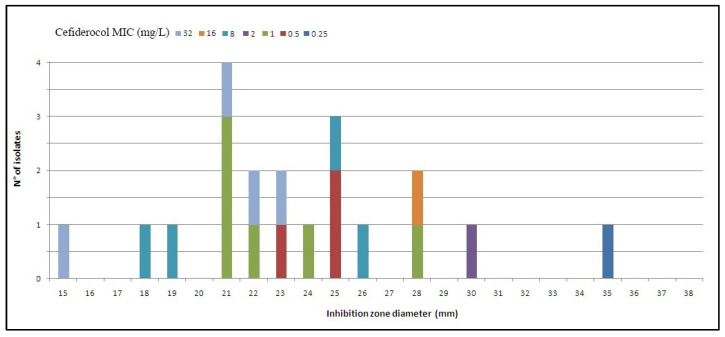
Cefiderocol 30 µg vs. MIC *Acinetobacter baumannii*, 20 isolates (EUCAST related breakpoints, MIC: S ≤ 2 mg/L; Zone diameter: S ≥ 17 mm [19]).

**Figure 2 antibiotics-10-01309-f002:**
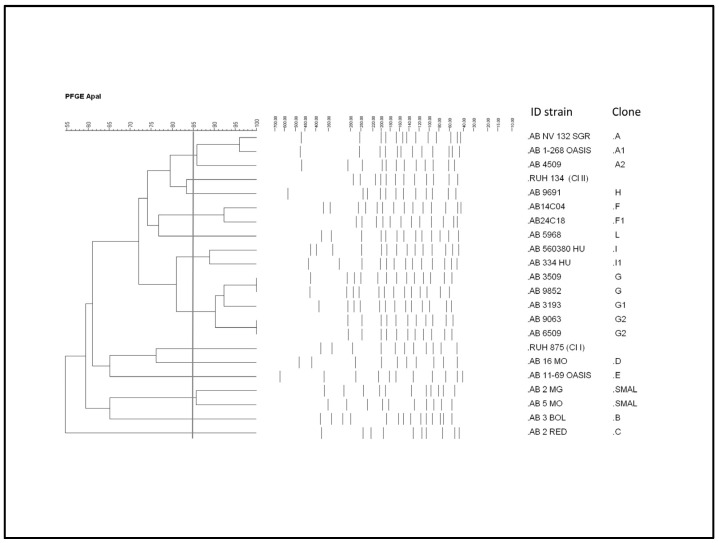
Cluster analysis of the pulsed-field gel electrophoresis profiles of the *Acinatobacter baumannii* strains included in the study.

**Table 1 antibiotics-10-01309-t001:** MICs values (mg/L) of the antibiotics tested for the *Acinetobacter baumannii* isolates *.

Isolates (Reference Number)	Source	OXA-23 K-SeT^®^	TZP	CIP	SXT	AN	LEV	GEM	MEM	CZA	CAZ	FDC (mg/L)	FDC Disk (mm)
AB 1–268 OASIS	Tracheal swab	Negative	>128	>1	>8	8	4	>8	32	>64	>64	1	22
AB 11–69 OASIS	Bronchoaspirate	Negative	32	>1	>8	>16	8	8	32	>64	>64	1	21
AB NV 132 SGR	Bronchoaspirate	Negative	>128	>1	>8	>16	8	>8	16	>64	>64	8	19
AB 2 RED	Bronchoaspirate	Negative	>128	>1	4	>16	8	>8	16	8	32	0.5	25
AB 2 MG	Urine	Negative	64	>1	4	16	8	>8	0.25	16	32	1	24
AB 3 BOL	Tracheal swab	Negative	>128	>1	>8	>16	8	>8	8	16	16	1	21
AB 24C18	Tracheal swab	Negative	16	>1	>8	16	>8	8	16	64	>64	8	18
AB 5 MO	Bronchoaspirate	Positive	32	1	8	>16	>8	>8	16	4	32	1	28
AB 560,380 HU	Urine	Positive	>128	>1	>8	>16	>8	>8	64	8	64	16	28
AB 9691	Wound	Positive	>128	>1	>8	>16	>8	>8	>64	64	>64	1	21
AB 14C04	Tracheal swab	Positive	≤2	0.06	≤4	≤2	1	≤1	4	2	2	0.25	36
AB 5968	Blood culture	Positive	>128	>1	>8	>16	>8	>8	>64	>64	>64	2	30
AB 9852	Blood culture	Positive	>128	>1	>8	>16	>8	>8	64	8	2	0.5	25
AB 3509	Urine	Positive	>128	>1	>8	>16	>8	>8	64	64	>64	0.5	23
AB 6509	Bronchoaspirate	Positive	>128	>1	>8	>16	>8	>8	64	64	64	8	26
AB 3193	Peritoneal fluid	Positive	>128	>1	>8	>16	>8	>8	>64	16	8	8	25
AB 334 HU	Bronchoaspirate	Positive	128	>1	>8	>16	>8	>8	32	>64	>64	>32	15
AB 16 MO	Tracheal swab	Positive	>128	>1	>8	>16	8	>8	64	32	64	>32	23
AB 9063	Peritoneal fluid	Positive	>128	>1	>8	>16	>8	>8	>64	>64	>64	>32	22
AB 4509	Urine	Positive	>128	>1	>8	>16	>8	>8	>64	>64	64	>32	21

* See Materials and methods for details. Broth microdilution panels: TZP: piperacillin/tazobactam; CIP: ciprofloxacin; SXT: trimetoprim-sulfametoxazol; AN:amikacin; LEV: levofloxacin; GEM: gentamicin; MEM: meropenem; CZA: ceftazidime/avibactam; CAZ: ceftolozane/tazobactam; FDC: cefiderocol. FDC disk: disk diffusion method using 30 µg discs of Cefiderocol.

**Table 2 antibiotics-10-01309-t002:** Molecular characterization of *Acinetobacter baumannii* isolates.

Isolates (Reference Number)	Resistance Determinants ^a^	Clone	MLST ^b^
AB 1–268 OASIS	*bla*OXA-58, ISAba1-*bla*OXA-51	A1	4
AB 11–69 OASIS	ISAba1-*bla*OXA-51	E	109
AB NV 132 SGR	*bla*OXA-58, ISAba1-*bla*OXA-51, *bla*OXA-11	A	new
AB 2 RED	*bla*OXA-58, ISAba1-*bla*OXA-51	C	78
AB 2 MG	ISAba1-*bla*OXA-51	SMAL	78
AB 3 BOL	*bla*OXA-58, *bla*OXA-128, ISAba1-*bla*OXA-51, *aphA6*	B	10
AB 24C18	ISAba1-*bla*OXA-51, *aphA6*	F1	2
AB 5 MO	*bla*OXA-23, ISAba1-*bla*OXA-51, ISAba1-*bla*OXA-23	SMAL	1077
AB 560380 HU	*bla*OXA-23, ISAba1-*bla*OXA-51, ISAba1-*bla*OXA-23, *armA*	I	2
AB 9691	*bla*OXA-23, ISAba1-*bla*OXA-51, ISAba1-*bla*OXA-23, *aphA6*	H	2
AB 14C04	*bla*OXA-23, ISAba1-*bla*OXA-51, *armA*	F	2
AB 5968	*bla*OXA-23, ISAba1-*bla*OXA-51, *armA, aphA6*	L	261
AB 9852	*bla*OXA-23, ISAba1-*bla*OXA-51, ISAba1-*bla*OXA-23, *aphA6*	G	2
AB 3509	*bla*OXA-23, ISAba1-*bla*OXA-51, *armA*	G	2
AB 6509	*bla*OXA-23, ISAba1-*bla*OXA-51, *armA*	G2	- ^c^
AB 3193	*bla*OXA-23, ISAba1-blaOXA-51, *armA*	G1	- ^c^
AB 334 HU	*bla*OXA-23, ISAba1-*bla*OXA-51, ISAba1-*bla*OXA-23, *aphA6*	I1	2
AB 16 MO	*bla*OXA-23, ISAba1-*bla*OXA-51, ISAba1-*bla*OXA-23, *aphA6*	D	19
AB 9063	*bla*OXA-23, ISAba1-*bla*OXA-51, *armA*	G2	2
AB 4509	*bla*OXA-23, ISAba1-*bla*OXA-51, *armA*	A2	new

^a^: The presence of constitutive *bla*OXA-51 determinant was not reported; ^b^: MLST: Multilocus sequence typing; ^c^: Not performed.

## Data Availability

The data presented in this study are available on request from the corresponding author.

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
