# Peer review of "In Vitro Antimicrobial Activity of the Siderophore Cephalosporin Cefiderocol against Acinetobacter baumannii Strains Recovered from Clinical Samples"

_antibiotics, 2021, doi:10.3390/antibiotics10111309_

Round 1
Reviewer 1 Report
This is a study of the antibacterial activity of cefiderocol against Acinetobacter baumannii strains from clinical samples. It is known that this siderophore cephalosporins exhibits antimicrobial activity against most multi drug resistant G- bacteria, therefore it is not very surprising that it also exhibits antibacterial activity against A. baumannii strains.
Nevertheless, it is of interest to study whether this is also the case with clinical samples. Maybe the most important finding is that large variability and low reproducibility was found. This is often the case with clinical samples, and should not be considered a weakness of the study. However, this referee would have liked a greater emphasis on this in the discussion of the methods and the results. Furthermore, I think it is to weak to conclude that further clinical studies are necessary without pointing out possible improvements or suggestions for further work.
Author Response
Reviewer 1:
Comments and Suggestions for Authors
This is a study of the antibacterial activity of cefiderocol against Acinetobacter baumannii strains from clinical samples. It is known that this siderophore cephalosporins exhibits antimicrobial activity against most multi drug resistant G- bacteria, therefore it is not very surprising that it also exhibits antibacterial activity against A. baumannii strains.
Nevertheless, it is of interest to study whether this is also the case with clinical samples. Maybe the most important finding is that large variability and low reproducibility was found. This is often the case with clinical samples, and should not be considered a weakness of the study. However, this referee would have liked a greater emphasis on this in the discussion of the methods and the results.
As suggested, we thank the reviewer and we have changed the manuscript accordingly: “Recently, Morris and coauthors compared cefiderocol disk diffusion method to broth dilution on carbapenem-resistant Enterobacterales and non-glucose-fermenting Gram-negative bacilli. They observed that disk diffusion is a convenient alternative approach to broth dilution for cefiderocol antimicrobial susceptibility testing, except for A. baumannii complex isolates [30]. Moreover, Albano and coauthors determined the MICs of cefiderocol for 610 Gram-negative bacilli. Broth dilution and agar dilution methods were used. The results showed significant discordance between agar dilution and broth dilution. Again, broth dilution proved to be the most reliable method for determining cefiderocol MICs [31].”
Furthermore, I think it is to weak to conclude that further clinical studies are necessary without pointing out possible improvements or suggestions for further work.
As suggested, we thank the reviewer and we have changed the manuscript accordingly: “Cefiderocol demonstrated in vitro antimicrobial activity against both carbapenem-susceptible and non-susceptible A. baumannii strains, regardless of both the type of carbapenemase present and clonality. The cefiderocol resistance observed in our analysis, might be associated with reduced, downregulated or absent expression of siderophore receptors, which allow the entry of antibiotics into the bacterial cell. Molecular analysis of genes encoding for siderophore receptors are certainly needed to investigate these drug resistance mechanisms, to better define the role of cefiderocol for the treatment of MDR A. baumannii infections. In addition, it could be useful to perform broth dilution assays in the presence of different iron concentrations, with the aim to observe the performance of cefiderocol activity in a simulated host system.

Reviewer 2 Report
Comments and Suggestions for Authors
The manuscript titled “In vitro antimicrobial activity of the siderophore cephalosporin cefiderocol against Acinetobacter baumannii strains recovered from clinical samples” lays out a very interesting description between multidrug-resistant A. baumannii strains isolates from clinical specimens (urine, bronchoaspirate, tracheal swabs) in the Microbiology Laboratory of Desio Hospital, Italy. Characterization of multidrug-resistant A. baumannii strains from local community is necessary to understand if antimicrobial activity of cefiderecol can control this opportunistic pathogen to improve the management of infections caused by this microorganism. The work is well performed and easy to follow. I think that this adds to the body of knowledge and recommend for acceptance with minor comments addressed below:
Please see below for individual points.
1.- Abstract: I suggest eliminate the text in lines 38-39, without data or results for expression of siderophore receptors is difficult some interpretations.
2.- In results and discussion, please add some lines to understand the reason to detect the ISAba1 genetic element and for interpretation of antibiotics tested. Also, I suggest to show in tables and figures which strains were from urine, bronchoaspirate, tracheal swabs, as is indicated in methodology to understand the clonal relatedness and typing.
3.- Discussion: lines 245-247. Description of 20 strains is a very important step to understand some dynamics of local epidemiology in Italy, so that I suggest to change the text to show the importance of your study and eliminate the line 245.
4.- In Discussion and conclusions: Please describe the importance of molecular characterization of clonal heterogeneity in the isolates.
Author Response
Reviewer 2:
Comments and Suggestions for Authors
The manuscript titled “In vitro antimicrobial activity of the siderophore cephalosporin cefiderocol against Acinetobacter baumannii strains recovered from clinical samples'' lays out a very interesting description between multidrug-resistant A. baumannii strains isolates from clinical specimens (urine, bronchoaspirate, tracheal swabs) in the Microbiology Laboratory of Desio Hospital, Italy. Characterization of multidrug-resistant A. baumannii strains from the local community is necessary to understand if antimicrobial activity of cefiderecol can control this opportunistic pathogen to improve the management of infections caused by this microorganism. The work is well performed and easy to follow. I think that this adds to the body of knowledge and recommend for acceptance with minor comments addressed below:
Please see below for individual points.
1.- Abstract: I suggest eliminate the text in lines 38-39, without data or results for expression of siderophore receptors is difficult some interpretations. As suggested, we thank the reviewer and we have changed the manuscript accordingly.
2.- In results and discussion, please add some lines to understand the reason to detect the ISAba1 genetic element and for interpretation of antibiotics tested. As suggested, we thank the reviewer and we have changed the manuscript accordingly: “The location of ISAba1 sequence in the upstream regions of these resistance genes may facilitate their overexpression thus increasing the resistance level to carbapenems.”
Also, I suggest to show in tables and figures which strains were from urine, bronchoaspirate, tracheal swabs, as is indicated in methodology to understand the clonal relatedness and typing. As suggested, we thank the reviewer and we have changed Table 1 accordingly.
3.- Discussion: lines 245-247. Description of 20 strains is a very important step to understand some dynamics of local epidemiology in Italy, so that I suggest to change the text to show the importance of your study and eliminate the line 245. As suggested, we thank the reviewer and we have changed the manuscript accordingly.
4.- In Discussion and conclusions: Please describe the importance of molecular characterization of clonal heterogeneity in the isolates. As suggested, we thank the reviewer and we have changed the manuscript accordingly: “Cefiderocol demonstrated in vitro antimicrobial activity against both carbapenem-susceptible and non-susceptible A. baumannii strains, regardless of both the type of carbapenemase present and clonality.”
